# RETHINKING TEXTURE PATTERNS IN TRANSFORMER NEURAL NETWORK FOR MEDICAL IMAGE ANALYSIS

## ABSTRACT

Lesion identification has been known as a major purpose in computer-aided diagnosis (CADs) and one of key tasks in radiomics. This study aims to explore the potential of transformer neural network by introducing texture patterns and features to tune the learning model for lesion differentiation from the benign tissues. We proposed texture transformer network (TxTN) by integrating three texture layers in vision transformer (ViT) to enhance the discriminative capability for medical image analysis. This inspirational idea is stemmed from one important insight into the architecture of ViT and its major shortcomings including topological destruction, the loss of geometrical information and the lack of global characteristics. By considering the definition and the property of image texture, ViT and texture pattern have a strong complementary relevance since the locality and globality are two basic requirements of the latter. Moreover, many well-known texture patterns have very good embeddability in attention mechanism since they are always represented by vectors or matrix, such as gray level co-occurrence matrix (GLCM)and histogram. Hereafter, we figured out a practical way to combine them by developing a pattern family layer, a texture presentation layer and a texture feature layer to embed into transformer network as the substitute of the pixel projection layer which is the major stem of above drawbacks. Their combinations not only make full use of advantages of texture and ViT but also have strong potentials to tune the deep learning models by mining more heterogeneous properties from patterns instead of pixels in various imaging modalities. Therefore, many texture patterns could be re-used in our approach, such as gray level co-occurrence matrix (GLCM), vector quantization (VQ), and so on. In the current study, our approach selected three texture patterns into TxTN, i.e. GLCM, VQ and Laplacian. To evaluate the effectiveness of our approach, our approach is finally testified over two public medical datasets and demonstrated very striking performances.

## 1 INTRODUCTION

Due to the breakthrough of deep learning in the past one decade, numerous remarkable successes have been witnessed in the field of medical image analysis especially for tumor identification which is one of key tasks for both Computer-aided diagnosis (CAD) and Radiomics Gillies et al. (2016); Yassin et al. (2018). However, most of the investigations were carried out with convolutional neural network (CNN) models which depend on very deep layers, complex architectures, and a massive number of parameters to mine and extract more salient features as the presentation of the object. Recently, a great deal of effort has been made to integrate the attention mechanisms in deep learning architectures. These attention-based models have become a hot topics due to their great ability to encode long-range dependencies and easily learn highly effective characteristics Vaswani et al. (2017); Zheng et al. (2017). Many relevant approaches have been put into applications in various medical imaging tasks Gu et al. (2020); Li et al. (2020). However, one critical issue of CNN should be the significant decrease in interpretability of visual features in the learning procedure which have led to lots of silent complaints about black-box operations of neural networks Wang et al. (2021). The importance of explicit handcrafted texture features in medical imaging are almost diminished since they have been replaced by the convolution operations in CNNs which seems simple but bring the great challenge of interpretability for experts in the field of medicine Tan et al. (2019). In various medical imaging modalities such as CT, MRI and ultrasound, texture involves analyzing tumor/lesion

properties such as roughness, coarseness, smoothness and regularity Nailon (2010). It can provide valuable insights and assist in differentiating various tissue types.

Recently, visual transformer (ViT) has shown many better advantages over CNNs such as less parameter, simple architectures and only fewer layers Han et al. (2022). Moreover, the attention mechanism within ViT has great potential to replace the standard convolution operations by operating on a sequences of image patches to capture texture features which can then be used in conjunction with traditional texture analysis methods to improve accuracy and interpretability Xu & Loy (2021). Nevertheless, we find there are some drawbacks and limitations. Firstly, it extracts texture information by reshaping 2-dimensional image blocks into a 1-dimensional vector for encoding which destroy the topological and geometrical structures in the original image. Secondly, it mostly relies on the local characteristics more than global features which does not fully utilize the texture information since it disregards the spatial arrangement and relationships between neighboring pixels. Unlike CNNs, which inherently possess spatial invariance due to their local receptive fields and shared weight parameters Howard et al. (2017), ViT processes each image patch independently Dosovitskiy et al. (2020). As a result, it may struggle with capturing local spatial relationships, especially when dealing with images containing objects at different scales or with intricate spatial structures. Additional techniques like patch overlapping or hierarchical architectures can partially mitigate this issue but may introduce additional computational overhead. How to combine the traditional texture patterns to tune the transformer model for better performances is another challenge.

To address these concerns, we proposed the Texture Transformer Network (TxTN), which merges various texture patterns and features with the Vision Transformer (ViT). This innovative approach aims to tune the learning model and unlock the substantial potential inherent in texture patterns within the context of a transformer neural network. This concept has been inspired by a pivotal insight into the ViT architecture, where we replace the pixel projection layer with three texture related layers. The texture patterns we incorporate encompass both local and global texture information, endowing TxTN with robust capabilities to optimize the model with a comprehensive array of texture features. Moreover, since all local features encompass not only the focal pixel but also its neighboring pixels, TxTN effectively preserves topological information and geometric structures. Several significant contributions of our approach can be outlined as follows: 1) introducing a novel method that seamlessly integrates ViT with texture patterns. As far as our knowledge extends, our work stands as the pioneering endeavor to tune transformer neural networks using texture patterns; 2) developing three innovative layers: the Pattern Family Layer, the Texture Representation Layer, and the Texture Feature Layer, all of which are seamlessly integrated into the transformer architecture. 3) proposing two GPU algorithms tailored for the calculation of texture patterns in the tensor domain, enhancing efficiency and scalability in this critical aspect of our approach.

## 2 RELATED WORKS

Radiomics represents a comprehensive approach, extending methods of computer vision into the domain of medical imaging. Texture plays a critical role in the analysis of medical images, encompassing tasks like segmentation, tissue detection, feature representation, and tissue classification Lambin et al. (2012). Given our research aims, we place particular emphasis on the intricacies of texture extraction, representation, and classification. Traditionally, texture extraction seeks to mine salient properties encompassing both micro- and macro- structural elements within textures, forming its fundamental requirement Tuceryan & Jain (1993). Notable outcomes of texture extraction manifest as texture patterns, including GLCM, GLDM, GLRLM, GLSZM, GLTDM, LBP, LTP, and more Strasburger et al. (2011). These patterns are considered as intermediate representations of texture, with their ultimate manifestation being a quantitative expression for textures. These expressions, often termed as texture features, are derived from these patterns or the original image using feature extraction methods like geometric approaches, statistical techniques, Fourier analysis, Gabor analysis, wavelet analysis, and more Zhang & Tan (2002). In radiomics, texture classification consistently employs renowned machine learning classifiers such as random forests, support vector machines (SVM), boosting, bagging, Bayesian methods, linear and non-linear regressions, K-means, LLE, IsoMap, and so on Leger et al. (2017). Each facet of the radiomics process offers a natural, lucid, and controllable framework for radiologists and experts, underpinning its widespread popularity within the field Parekh & Jacobs (2016).

The advent and widespread use of deep learning architectures, particularly CNN and ViT, have indeed ushered in a transformative era in image analysis, revolutionizing traditional radiomics procedures Avanzo et al. (2020); Chetoui & Akhloufi (2022). As is well-known, both CNN and ViT are end-to-end systems that have reshaped and replaced conventional texture processing techniques with operations like convolution, pooling, activation, attention, and traditional neural networks Howard et al. (2017); Dosovitskiy et al. (2020). These operations aim to mimic the iterative learning process of human cognition, albeit in a somewhat opaque manner, leading to the perception of "the black box" among many radiologists and researchers Castelvecchi (2016). However, with further studying, we discover that there are still underlying similarities between traditional radiomics and deep learning methods. Deep learning methods continue to follow some of the fundamental concepts employed in radiomics. Firstly, the extraction and utilization of texture features still persist in deep learning approaches. CNNs employ multiple layers that combine convolutional operations, pooling methods, activation functions, and flattening operations to generate global image features. Similarly, visual transformers utilize patch subdivision, attention mechanisms, normalization, and activation functions to produce features. Secondly, the concept of feature selection is also present in both CNN and ViT architectures. They employ parameter weights to iteratively prioritize important features while disregarding redundant information. Thirdly, the final step in both traditional radiomics and deep learning involves the use of a classifier. In this regard, both CNN and ViT architectures utilize neural networks as classifiers to train models based on the extracted features Alzubaidi et al. (2021). In essence, while deep learning methods introduce complexity and opaqueness with their advanced architectures, they still maintain a connection to the fundamental principles that underpin traditional radiomics procedures.

Rather than CNN, visual transformers introduced attention mechanism instead of the convolution operation and pooling to produce texture features Dosovitskiy et al. (2020); Liu et al. (2021) It is a groundbreaking architecture introduced in the paper "An Image is Worth 16x16 Words: Transformers for Image Recognition at Scalev" by Dosovitskiy et al. Dosovitskiy et al. (2020). It herits the main idea of the method in natural language processing (NLP) and has since been widely adopted in various domains, including computer vision and reinforcement learning Han et al. (2022) The Transformer model relies heavily on the self-attention mechanism to capture global dependencies in the input data without the need for recurrent or convolutional operations. ViT architecture contains some key components: 1) Encoder-Decoder Architecture, 2) Positional Encoding, and 3) Multi-Head Attention Dosovitskiy et al. (2020); Liu et al. (2021). By Comparison with CNN, it has shown very great advantages to overcome the memory occupitation and time cost in model training Touvron et al. (2021). Some research with transformers on medical image analysis have emerged since 2021. Wu et al. (2023) put SWIN Transformer into hepatic vessel analysis for its segmentation. Du et al. (2022) proposed SWIN Transformer-based multiscale feature pyramid aggregation network for medical image segmentation. Huang et al. (2022) introduced Scale-former for multi-task medical image segmentation which utilizes transformer networks as the backbone. Yu et al. (2023) proposed local spatial representation learning with hierarchical transformer for efficient medical segmentation. Nalawade et al. (2021) introduced Transformer networks to create a federated learning for brain segmentation. Jang & Hwang (2022) proposed three-dimensional medical image classifier using multi-plane and multi-slice transformer to carry out brain image classification. Wang et al. (2022) utilized a Transformer model to explore functional near-infrared spectroscopy classification. Li et al. (2022) processed EEG-based emotion recognition via transformer neural networks. Moreover, Transformer networks have been also applied to conduct image analysis and medical image registration Sarasua et al. (2022); Xie et al. (2022). Some other applications in medical imaging have been reported by two reviewing works Parvaiz et al. (2023); Shamshad et al. (2023).

In medical images, texture patterns provide important cues for understanding and analyzing visual information of tumors or lesions Zhang & Tan (2002). Hence, its combinations with CNN have been explored by GLCM-CNN method and TPPNet Methods and achieve very striking outcomes Tan et al. (2019); Cao et al. (2023). The former approach integrated the pattern of GLCM with CNN architecture to carry out polyp classification over a colonography dataset while the later designed a family of triple point patterns (TPP) which are fed into a simplified CNN architecture to conduct abnormal brachial plexus (BP) differentiation from the normal BP MRIs. Both approaches have proved the practicability of texture pattern in deep learning. This article aims to carry out some investigations on the combination of texture patterns and transformer neural networks.

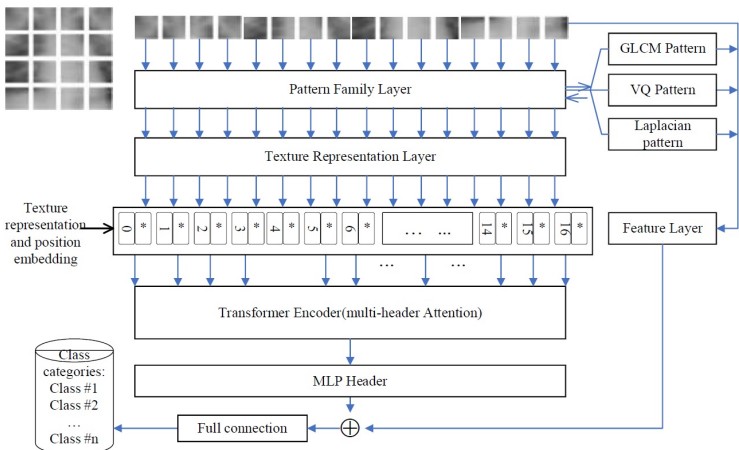

Figure 1: Texture transformer architecture where $\bigoplus$ denotes vector concatenation.

# 3 METHOD

## 3.1 TEXTURE TRANSFORMER

In this section, we introduce a comprehensive framework encompassing the development of three key components: **pattern factory layer**, **texture representation layer**, and **texture feature layer** . These components are designed to seamlessly integrate into the Vision Transformer (ViT) architecture, effectively replacing the traditional pixel mapping layer. As depicted in Figure 1, the pattern factory layer is primarily responsible for extracting texture patterns from image patches. This layer serves as an abstract module, effectively acting as a container for creating various texture pattern families. Next, the texture representation layer takes the extracted texture patterns and processes them to generate meaningful representations of the image patches. This step functions as a converter, transforming the extracted texture patterns into vector representations with a uniform shape. The texture feature layer, on the other hand, is geared towards extracting global features from the tissue images. This layer leverages traditional feature extraction methods, capitalizing on well-established techniques to capture essential characteristics of the tissue. It attempts to re-use traditional features extracted from both the original images and the texture patterns.

Our proposed texture transformer provides a wonderful solution for two major concerns of ViT methodologically by introducing three layers, i.e. pattern family layer, texture representation layer and texture feature layer. Firstly, we introduced texture patterns to address the topological and geometrical structure issue since many texture patterns have strong capability to extract and maintain both topological and geometrical information locally and globally. This is the fundamental requirements of image textures Guo et al. (2007). Secondly, many texture patterns are always expressed by vectors and matrices which have very strong potential to embed into transformer neural network to encode with the attention mechanism to produce some sophisticated high level features. Additionally, the attention module could directly study features from texture patterns instead of low-level pixels which could tune and guide our model to carry out some special tasks in medical image learning. Moreover, we also add the texture feature layer because we find some tissue features, such as surface area, volume, skewness, kurtosis, compactness, geodesic length, and so on, still play very critical roles in radiomics and CAD. However, it is not easy to extract these geometrical measures with deep learning. These features could be re-used as the input of the full connection module by combining with the high-level features extracted with deep learning.

## 3.2 PATTERN FACTORY LAYERS

Over the past three decades, numerous texture pattern families have been developed Zhang & Tan (2002). To ensure consistency in the input format for the transformer encoder, we have purposefully selected several patterns that can be represented as vectors or matrices. In this article, we focus on three specific patterns and their fast algorithms on GPU, which are detailed in this section.

### 3.2.1 LAPLACIAN PATTERN

A Laplacian pattern is designed to extract local texture patterns via Laplacian operator which is defined by two second order derivatives and has widely used in image processingHao et al. (2023). its definition in 2-dimensional Euclidean space ($\Re$) is give by

$$\triangle I = I_{xx} + I_{yy} = \frac{\partial^2 I}{\partial x^2} + \frac{\partial^2 I}{\partial y^2}$$

where $I$ is a 2-dimensional image, $x, y \in \Re$. On 2-dimensional digital images, the Laplacian could be replace with two filters as the following Paris et al. (2011)

$$f_x = \begin{bmatrix} 1 & -2 & 1 \\ 2 & -4 & 2 \\ 1 & -2 & 1 \end{bmatrix}, f_y = \begin{bmatrix} 1 & 2 & 1 \\ -2 & -4 & -2 \\ 1 & 2 & 1 \end{bmatrix} \tag{1}$$

In $Eq(1)$, it is demonstrated that the Laplacian operator can be regarded as a representation of local patterns because it incorporates information from all of its 1-ring neighbors during computation. In our paper, we adopt the magnitude of the Laplacian, denoted as $\sqrt{(I_{xx})^2 + (I_{yy})^2}$, as a means to characterize these local patterns. The calculation of this magnitude reveals that the Laplacian not only extracts local texture details but also maintains certain aspects of local topological structures within the image. To address any concerns about its global behavior, we employ statistical methods such as histogram analysis, which helps in capturing and representing the overall distribution of these local patterns across the entire image. We designed a GPU algorithm for histogram calculation as shown in the Histogram_GPU as the following where the $stack, tile, transpose$ functions are commonly used in libraries of $Numpy$ and $Tensorflow$.

---

**Algorithm Histogram_GPU**

---

**Require**: T deotes a 2D tensor with the size of [batch*patches, patch_size*patch_size], nbins=m denotes the image gray level

  1 **Function** Histogram_GPU
  2 row,col=T.shape
  3 Pools=create_m_histogram_pools_according_to_image_level() # its size is m*m
  4 pT=stack(pools)
  5 nT=tile(T,[nbins,row,col])
  6 histogram=trranspose(equal(pT, nT), axis=2)
  7 return histogram

---

### 3.2.2 GLCM PATTERN

GLCM Pattern serves the purpose of extracting texture patterns, which are essentially 2-dimensional distributions of pixel-pairs within an image. These pixel-pairs are instrumental in defining a local measure that represents their topological relationship, encapsulating highly effective local properties. Simultaneously, the distribution derived from the GLCM denotes global features, illustrating the ratio of each local measure in the context of the entire image. The formal definition of the GLCM can be found in reference Cao et al. (2021)

$$C(i, j, d, \theta) = \sum_{p \in D} \begin{cases} 1 & I(p) = i, I(p + d * (cos\theta, sin\theta)) = j \\ 0 & otherwise \end{cases} \tag{2}$$

where $I$ denotes the image, $p$ is a point within the image domain ($D$), $I(p)$ is a pixel at point $p$ in $I$, $d$ is a shifting distance from point $p$, $(i, j)$ indicates an image pixel pair, and $\theta$ shows the direction (angle) of the shifting distance. In our practice, the $d$ is set to 1. Totally, there are four angels, i.e. $0^o, 45^o, 90^o and 135^o$. In digital images, the shift operation could be substituted with a shifting filter. Shift filters of $0^o, 45^o, 90^o and 135^o$ are given by

$$s_0 = \begin{bmatrix} 0 & 0 & 0 \\ 1 & 0 & 0 \\ 0 & 0 & 0 \end{bmatrix}, s_{45} = \begin{bmatrix} 1 & 0 & 0 \\ 0 & 0 & 0 \\ 0 & 0 & 0 \end{bmatrix}, s_{90} = \begin{bmatrix} 0 & 1 & 0 \\ 0 & 0 & 0 \\ 0 & 0 & 0 \end{bmatrix}, s_{135} = \begin{bmatrix} 0 & 0 & 1 \\ 0 & 0 & 0 \\ 0 & 0 & 0 \end{bmatrix} \tag{3}$$

To accelerate the computation of GLCM on a GPU within the TensorFlow environment, we devised a high-speed GLCM computation method, documented as the GLCM_GPU algorithm. Within this layer, a multitude of 2-D matrices are swiftly generated and subsequently serve as inputs for the subsequent layer. These matrices facilitate the extraction of elevated-level features for each lesion, enhancing the efficiency and effectiveness of the overall process.

---

**Algorithm** GLCM_GPU

---

**Require**: P deotes image patches with the size of [batch, patches, patch_size, patch_size, channel], f denotes shift filter with the size of [3,3],nbins=m represents the image gray level

1 **Function** GLCM_GPU
2 Pg=(P-min(P))/(max(P)-min(P))*(nbins-1)
3 Ps=reshape(Pg, [batch*patches,patch_size,patch_size,channel]
4 Pc=reshape(Ps, [batch*patches,patch_size**2,channel])
5 Pf=conv2d(Pg,f)
6 Pf=reshape(Pf, [batch*patches,patch_size**2,channel])
7 Pa=Pf*nbins+Pc
8 Ph=histogram_patch(Pa)
9 Pd=reshape(Ph,[batch,patches,nbins,nbins,channel]
10 return Pd

---

### 3.2.3 VQ Pattern

VQ (Vector Quantization) is introduced to leverage Vector Quantization techniques for mapping each image patch into a new eigen space. VQ is a well-established method employed in signal processing and data compression to reduce the data needed to represent a signal or an image Shlezinger et al. (2020). It achieves this by partitioning a set of continuous or discrete vectors into clusters and representing each vector with the index of the cluster to which it belongs. In the context of discrete images, the vector is typically constructed using a central pixel and its 1-ring neighbors, effectively preserving the topological relationships between each pixel and its neighbors. Assuming the size of the image $I$ is $M * N$, we can create a $[M * N, 9]$ matrix. Subsequently, a PCA (Principal Component Analysis) technique is applied to calculate its principal components, which consist of 9 eigenvalues and their corresponding 9 eigenvectors Daffertshofer et al. (2004). In our approach, the top 2 eigenvectors are employed to construct a new 2D diagonal space, which is used to map the $[M * N, 9]$ matrix into a $[M * N, 2]$ matrix. This mapping operation essentially serves as a space embedding technology, capable of capturing semantic and syntactic relationships between pixels. This mapping is advantageous for tuning the learning model and enhancing lesion identification. A visual representation of this procedure is depicted in Figure 2. We ultimately generate histograms of 2D vector magnitudes from each row of the $[M * N, 2]$ matrix. The approach for characterizing and calculating its representation follows the Histogram_GPU Algorithm for Laplacian pattern.

### 3.3 Texture Representation Layer

This layer serves as a shape normalization step, which is essential because the preceding layer generates texture patterns in either vector or matrix form. To maintain uniformity in the input format for the transformer encoder, it is necessary to convert each texture pattern into a vector, which will

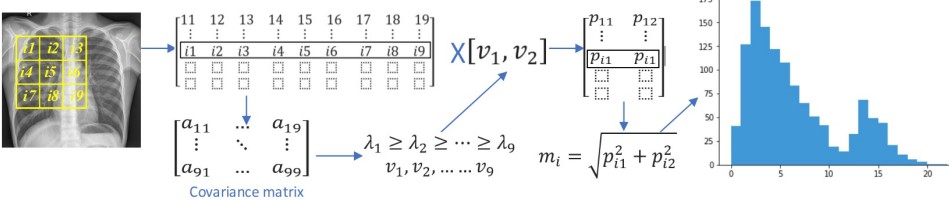

Figure 2: Procedure of VQ texture pattern where $\lambda_k$ and $v_k$ are k-th eigenvalue and eigenvector derived from co-variance matrix,$k \in \{1, 2, ..., 9\}$, $[p_{i1}, p_{i2}]$ are the projection of [i1,...,i9] in the eigen space with the basis of $v_1$ and $v_2$.

represent the texture presentation for each image patch. The conversion process depends on the nature of the texture pattern:

- If the texture pattern is already in vector form, it can be directly treated as the texture presentation.
- If the texture pattern is in matrix form, it can be mapped to a vector through a flattening method.

### 3.4 FEATURE LAYER

The Feature Layer is strategically designed to supplement the deep features extracted by the deep learning branch with traditional image features. Typically, two main categories of features are considered: texture features and shape features. In cases where no mask or Region of Interest (ROI) information is available, our focus is primarily on texture features Van Griethuysen et al. (2017). However, if mask or ROI information is accessible, our approach takes both texture and shape features into account. Shape features encompass critical metrics such as pixel surface, perimeter, sphericity, spherical disproportion, and elongation Van Griethuysen et al. (2017). This comprehensive consideration of both texture and shape features empowers our approach to provide a more holistic and informative feature set for the deep learning branch, thereby enhancing the model's capability for a wide range of image analysis tasks.

## 4 EXPERIMENT

### 4.1 DATASETS AND MATERIALS

Two medical image databases selected from MedMNIST v2 are adopted to testify our approach in the experiment.

**Pneumonia dataset** was constructed and released by Kermany et al. (2018) and was re-collected by MEDMNIST dataset Yang et al. (2021; 2023). According to the World Health Organization (WHO), approximate 2 million children under 5 years old every year will be killed by pneumonia which is consistently estimated as the single leading cause of childhood mortality, killing more children than HIV/AIDS, malaria, and measles combined. The two leading causes of pneumonia are bacterial and viral pathogens. Chest X-rays are routinely obtained as standard of care and can help differentiate between different types of pneumonia. This database collected and labeled a total of 5,232 chest X-ray images from children, including 1,349 normal and 3,883 pneumonia (2,538 bacterial and 1,345 viral), which are selected from a total of 5,856 patients.

**Breast dataset** Al-Dhabyani et al. (2020) consists of of 780 breast ultrasound images which were collected and stored in a DICOM format at Baheya Hospital, Egypt. It is categorized into 3 classes: normal, benign and malignant. All images were cropped to different sizes to remove unused and unimportant boundaries from the images with the resolution of 1×500×500. As only low-resolution images are used, the task was simplified into binary classification by combing normal and benign as positive, and classify them against malignant as negative.

All images with a window size of length of the short edge were center-cropped and resized them into 28x28 for both datasets. Moreover, 5-fold cross-validation scheme was introduced to generate 5 cohorts with a stochastic manner. Each cohort contains training, validation and testing subgroups with a ratio of 6:2:2.

### 4.2 IMPLEMENTATION DETAILS

Some specificities of our computing platform contains one AMD EPYC 7352 24-Core Processor, 1TB memory and a Nivida A100-SXM GPUs with 80GB GPU memory. The whole dataset was divided into three subsets according to their ratio mentioned above. Some critical parameters for the model training include the optimizer, learning rate, decay rate, patch number, projection number, attention head number, transformer unit, transformer layer number and mlp head units. we choose Adamw as the optimizer with learning 0.001 and decay rate 0.0001. All images are resized to 72*72. The projection dimension is 64. Attention head number and transformer layer are set to 4 and 8

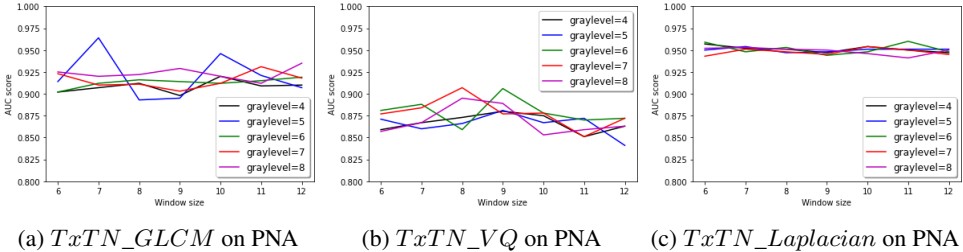

(a) $TxTN\_GLCM$ on PNA     (b) $TxTN\_VQ$ on PNA     (c) $TxTN\_Laplacian$ on PNA

Figure 3: Ablation study on the window size and graylevel over pneumonia (PNA) dataset evaluated by AUC score.

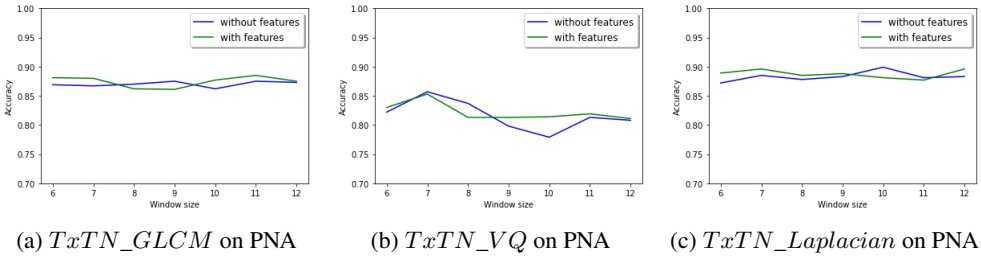

(a) $TxTN\_GLCM$ on PNA     (b) $TxTN\_VQ$ on PNA     (c) $TxTN\_Laplacian$ on PNA

Figure 4: Ablation study on the texture feature layer over pneumonia (PNA) dataset by adding and excluding shape features where the evaluation measure is accuracy.

respectively. The transformer unit is set to [128, 64] and the MLP head unit is [2048,1024]. The patch size is varying in [6,8,9,12]. The gray level is another important parameters. If it is very large, the GLCM and the histogram will be too sparse. If it is too small, both histogram and GLCM do not make any sense in statistics. Therefore, we empirically keep at least 6 hits for each bar in average. To evaluate TxTN's performance, we employ accuracy, AUC score (Area under the ROC Curve), F1-score as the evaluation measure. The epoch is set to 100 with the batch size of 256.

## 4.3 ABLATION STUDY

**The gray level** of the image and **the window size**(or the patch size) of TxTN are two important arguments in texture pattern construction. By considering the small size of image patches, we chose some small gray levels to test the performance of TxTN to reduce the side effect of sparse matrix(as the characterization of texture patterns). We use min-max approach to re-scale the image. The window size is varying from 6 to 12. meanwhile, the gray level is set in [4,5,6,7,8]. Figure 3 plots some AUC score curves over pneumonia datasets with various combinations of patch size and gray level. $TxTN\_GLCM$'s performances over the breast dataest are better than $TxTN\_VQ$ and $TxTN\_Laplacian$ while $TxTN\_Laplacian$ achieves the best performances over the pneumonia dataset. We also find that $TxTN\_lapalacian$ is relatively more stable than $TxTN\_GLCM$ and $TxTN\_VQ$ while changing gray level and window-size. The performances over the breast dataset is demonstrated in Appendix.

**Feature layer** also plays some critical roles in TxTN. Because there is no ROI, we only choose some texture features in our study. All these texture features are listed in Appendix.They are all complementary for the high level deep learning features. The lesion identification performances on the pneumonia dataset are demonstrated in Figure 4 which shows that these measures have potential to tune the model for lesion classification. The experimental outcomes over the breast dataset is demonstrated in Appendix.

## 4.4 COMPARISONS

Three state-of-the-art methods are employed to make some comparisons with our approach as follows:

- **MobileNet** is a well-known light-weighted CNN architecture Howard et al. (2017). It has demonstrated high efficiency and stability in many applications.
- **ViT** is the first paper to put the transformer network into computer vision and get high appraise in medical image analysis Dosovitskiy et al. (2020).
- **SWIN** is another transformer based neural network which utilizes a hierarchical structure and shifted windows to process image analysis Liu et al. (2021).

In model training, three of them adopt Adam optimizer with learning rate 0.001 and weight decay 0.0001. Their epoches and batch numbers are shared with our approach in Section 2.2. ViT adopts the same parameters as our TxTN. SWIN choose different parameters to train the model. Its patch size is set to 2. The head number and embed dimension, and MLP number are 8,64 and 256. The window size and the shift size are set to 2 and 1. The ratio of validation splitting over the pneumonia dataset is set 0.1. For our approaches, the graylevel of Laplacian layer and VQ layer shares 8 and their patch size is set to be 9. Meanwhile, their graylevels and batch sizes are set to 4 and 8 respectively. Three evaluation measures are given in Table 1 and their ROC curves are plotted in Appendix. Both Table 1 and Figure 5 exhibit the superiority of the proposed approaches compared to the state-of-the-art methods.

Table 1: Three evaluation measure comparisons among six approaches over the breast and pneumonia dataset.

| Method | Breast (Ultrasound) | | | Pneumonia(X-ray) | | |
|---|---|---|---|---|---|---|
| | AUC | ACC | F1-score | AUC | ACC | F1-score |
| MobileNet | 0.747 | 0.756 | 0.851 | 0.704 | 0.725 | 0.769 |
| SWIN | 0.762 | 0.708 | 0.845 | 0.923 | 0.865 | 0.896 |
| ViT | 0.734 | 0.782 | 0.864 | 0.917 | 0.867 | 0.899 |
| $TxTN\_GLCM$ | 0.812 | 0.827 | 0.889 | 0.959 | 0.910 | 0.920 |
| $TxTN\_Laplacian$ | 0.807 | 0.821 | 0.883 | 0.964 | 0.931 | 0.945 |
| $TxTN\_VQ$ | 0.817 | 0.833 | 0.894 | 0.907 | 0.852 | 0.887 |

## 5 CONCLUSION

In this article, we proposed a novel transformer neural network, termed as texture transformer (abbr TxTN), to combat some challenges, such as topological destruction, the loss of geometrical information and the lack of global characteristics, stemmed from the vision transformer by embedding the texture pattern layers, histogram layer and texture feature layer into the transformer network. It not only optimizes the pixel embedding layer but also successfully tune the learning model by involving more topological and texture information to mine more effective heterogeneous feature for several medical imaging modalities. To make full use of tensor based computation, we also develop two fast algorithms in GLCM and histogram calculation. Experimental outcomes on the pneumonia and breast datasets demonstrate the plausibility and effectiveness of our proposed approach. This paper presents our initial exploration into the texture and its integration with ViT. However, as we look ahead to future research, two impportant concerns come to the forefront. The first concern revolves around the need to delve deeper into the world of texture patterns for the Texture Transformer Network (TxTN). Expanding our repertoire of texture patterns will likely enrich the capabilities of TxTN and enhance its effectiveness. The second concern is centered on the challenge of seamlessly integrating multiple texture patterns into a cohesive texture representation within this transformer architecture. This integration task is crucial for achieving a more holistic understanding of texture and maximizing its potential impact within the framework of transformer-based models.

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

## A    APPENDIX

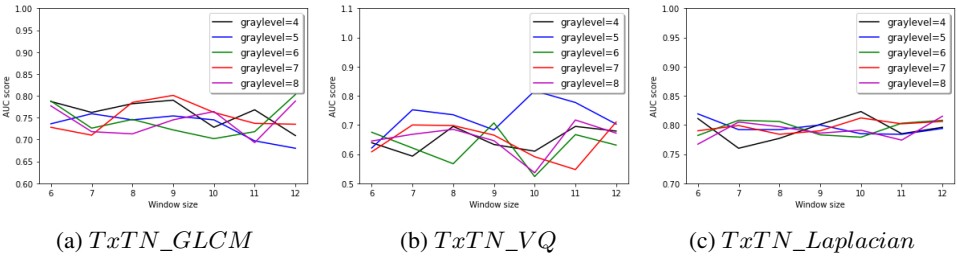

(a) $TxTN\_GLCM$        (b) $TxTN\_VQ$        (c) $TxTN\_Laplacian$

Figure 5: Ablation studies on window size and image graylevel of each patches over the breast dataset evaluated by AUC score where patch size= $window\_size * window\_size$.

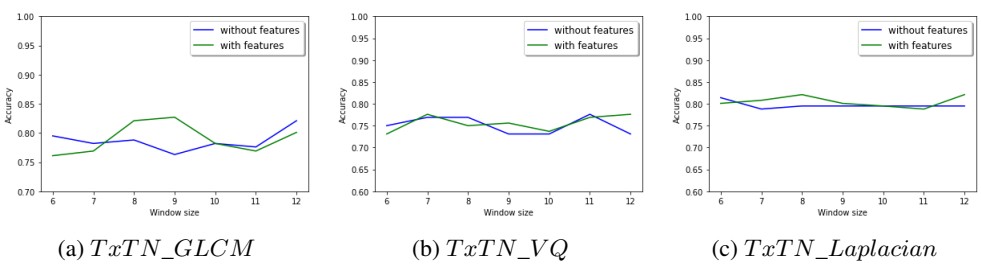

(a) $TxTN\_GLCM$        (b) $TxTN\_VQ$        (c) $TxTN\_Laplacian$

Figure 6: Ablation study on the texture feature layer over the breast dataset by adding and excluding shape features where the evaluation measure is accuracy.

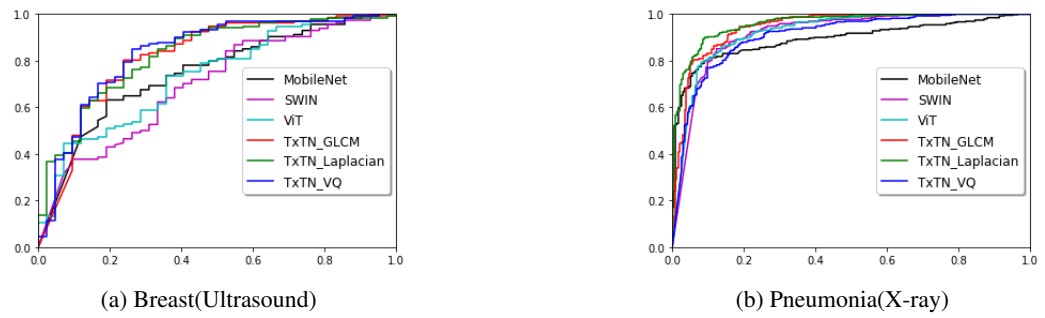

(a) Breast(Ultrasound)        (b) Pneumonia(X-ray)

Figure 7: ROC curves of six approaches over breast and pneumonia dataset.

Table 2: Texture features used in our ablation study Van Griethuysen et al. (2017).

| Feature | Texture Feature List |
| --- | --- |
| GLCM | Autocorrelation, ClusterProminence, ClusterShade, ClusterTendency, Contrast, Correlation, DifferenceAverage, DifferenceEntropy, DifferenceVariance, Id, Idm, Idmn, Idn, Imc1, Imc2, InverseVariance, JointAverage, JointEnergy, JointEntropy, MCC, MaximumProbability, SumAverage, SumEntropy, SumSquares |
| Histogram | Energy, Total Energy, Entropy, Minimum, 10th percentile, 90th percentile, Maximum, Mean, Interquartile Range, Range,Mean Absolute Deviation, Robust Mean Absolute Deviation, Root Mean Squared, Standard Deviation, Skewness, Kurtosis, Variance, Uniformity |

