# OpenReview forum: "Rethinking Texture Patterns in Transformer Neural NetWork for Medical Image Analysis"
_ICLR.cc/2024/Conference — Submitted to ICLR 2024_

### Official Review · Reviewer_M5DR · 2023-10-16

**Soundness:** 2 fair
**Presentation:** 2 fair
**Contribution:** 2 fair
**Rating:** 3
**Confidence:** 5

**Summary:**

The authors propose a transformer network that incorporates a module extracting texture features (Laplacian, Grey Level Occurrence Matrices, and VQ) in particular and evaluate it on binary classification tasks of two medical datasets of different body sites and modalities (ultrasound and X-ray).

**Strengths:**

It has been previously shown that fusing classic texture features and learned features, or putting a particular focus on texture extraction within networks by having texture-extracting modules can be beneficial for image classification in comparison to general-purpose CNNs, in particular in biomedical applications when little (labelled) training data is available. Hence, the idea of introducing texture features in the context of ViT is interesting and promising.

The authors consider two different data sets, in two different modalities, and different body sites, both are relatively big from a biomedical perspective, which is a strength of the paper.

The authors provide a sensitivity study on some hyperparameters of the texture features, and an ablation study of additionally including shape features in the training.
They compare their method with three baselines (MobileNet, SWIN and ViT).

**Weaknesses:**

Methods & Experiments
The description of the technical details how the texture features are extracted and used within the network is not completely clear to me. Which parameters within the texture layers are learned? How is their position embedded as described in Fig 1? Are the texture features concatenated again with the features from the transformer stream that also takes them as input (as Fig 1 suggests)? The description of the method provided in the manuscript would not be enough for me to reimplement the method.
Additionally, I am confused regarding the data preprocessing (see below).
Hence, I find it hard to assess the experiments and think the manuscript would benefit from a revision of the text.




Originality
The authors provide a lengthy background sections on the use CNNs and ViT in general and in medical applications, but only very little on work that combines texture descriptors and CNNs. Since this is a crucial part of introduced method, I suggest developing this part of the background section a bit more in trade-off with the more general background. Some of that work may include (follow-up work on this should be further checked and also considered to compare to as a baseline):
Detection of cervical cancer cells based on strong feature CNN-SVM network (Jia et al., 2020)
F. Juefei-Xu, V. N. Boddeti and M. Savvides, "Local Binary Convolutional Neural Networks", Proc. CVPR, 2017.
L. Li et al., "Face spoofing detection with local binary pattern network", J. Visual Commun. Image Represent.
G. Levi and T. Hassner, "Emotion recognition in the wild via convolutional neural networks and mapped binary patterns", Proc. of ACM Int. Conf. on Multimodal Interaction
“Learning Texture Transformer Network for Image Super-Resolution”, Fuzhi Yang, Huan Yang, Jianlong Fu, Hongtao Lu, Baining Guo; Proceedings of the IEEE/CVF Conference on Computer Vision and Pattern Recognition (CVPR), 2020

I also miss some references to the original work introducing the texture features used in this study, i.e. the Laplacian patterns provide a reference to Hao et al. from 2023, the GLCM patterns to Cao et al. 2021, etc. These authors may have used those patterns in their studies, but this should go into the relevant background section. These texture features have been introduced very long ago, e.g. GLCM in the early 70s in Haralick R. M., Shanmugam K. & Dinstein I. Textural Features for Image Classification. IEEE Transactions on Systems, Man, and Cybernetics 3, 610–621 (1973), which should be provided as a reference accordingly.


Clarity
Eqn (1): $\mathcal{R}$ is defined as a 2-dim Euclidean space just before the definition of the Laplace operator, but x,y are also in $\mathcal{R}$, which doesn’t work out and has to be adjusted. When revisiting this definition, mind that this is a discrete Laplace operator, i.e. x and y should be $\mathcal{N}_x, \mathcal{N}_y \subset \mathbb{N}$, and $I$ a function mapping from $\mathcal{N}_x \times \mathcal{N}_y$ to $[0,1]$, or whichever intensity range is chosen.

The manuscript would benefit greatly from some being reiterated by a native speaker or software to correct for some of the used grammar, punctuation, and choice of words. Overall it is clear enough to understand the authors' intentions and it is for the most part not ambiguous, but there are grammar mistakes and confusing phrases on many occasions.

There are many examples in the text, but just to pick one, the paragraph following Eqn (1):
“The calculation of this [Laplacian] magnitude reveals that the Laplacian not only extracts local texture details but also maintains certain aspects of local topological structures within the image. To address any concerns about its global behavior, we employ statistical methods such as histogram analysis, which helps in capturing and representing the overall distribution of these local patterns across the entire image”.
Sentences like these leave me wondering in which way the magnitude (or its calculation to be more precise) maintains aspects of local topological structures? How it addresses concerns about its global behavior? What are these concerns at all? What is the histogram analysis used? In what way is that a statistical method?

The paragraph on the VQ pattern is quite unclear. What is the 1-ring neighbor? I assume because the authors then talk about 9 principal components, that they mean a 3x3 neighborhood which they decompose, but it is not clear from the text why there are 9 eigenvalues/vectors.

From the paragraph on GLCM it’s not really possible to later conclude what the implementation details mean, when the authors say “	The gray level is another important parameters. If it is very large, the GLCM and the histogram will be too sparse. If it is too small, both histogram and GLCM do not make any sense in statistics. Therefore, we empirically keep at least 6 hits for each bar in average.” Please try to define the required parameters when introducing GLCM and use the same terminology throughout the paper.

Additionally, there are sometimes spaces missing, words repeating, text used in math environment.




-----------------------------------------------------------------------
After Author-Reviewer-Discussion:

I acknowledge having read through the other reviewers' reviews and remain with my initial score.

**Questions:**

Algorithm Histogram_GPU - what is the significance of this function? In which way is it GPU-specific?

I think all the algorithms provided in the manuscript could be moved to the appendix.

It comes as a bit of a surprise to read in 3.4 that in some cases also shape features are extracted and merged with the texture features. How are they extracted? How are they merged with other features during training?

What are the dimensions of the images in the pneumonia dataset?

How many images were there in each class in the breast dataset?

All images were resized to 28x28 pixels? In the implementation details it then says they were resized to 72x72? Please clarify.

What does “By considering the small size of image patches, we chose some small gray levels to test the performance of TxTN to reduce the side effect of sparse matrix(as the characterization of texture patterns).” mean?

In Fig. 1 it says “Texture Representation and position embedding”, how does the position embedding work?

---

### Official Review · Reviewer_ej1o · 2023-10-17

**Soundness:** 3 good
**Presentation:** 4 excellent
**Contribution:** 2 fair
**Rating:** 3
**Confidence:** 5

**Summary:**

This paper proposed an improved ViT model by introducing texture layers into attention parts for medical image classification. The texture layers are designed by standard GLGM, Laplacian modules. Experiments on two medical datasets demonstrate that paying more attention to texture features can improve classification performance than ViT models.

**Strengths:**

1. Good performance gain
2. Clear motivation and writing

**Weaknesses:**

1. Technical novelty is lacking. Transformer-based methods can better capture high-level information rather than textural features. These insights have been shown in previous works, see [1]. Here, CNN may be able to learn the textural information well, see [2]. Therefore, why not combine both for medical tasks? Furthermore, the comparison is not sufficient. there are many customized methods for medical tasks, which should be included for the comparison. Finally, do the fixed GLGM, Laplacian modules denote all textural features? More analyses are needed here.
[1] Ghiasi, Amin, et al. "What do vision transformers learn? a visual exploration." arXiv preprint arXiv:2212.06727 (2022)
[2] Geirhos R, Rubisch P, Michaelis C, et al. ImageNet-trained CNNs are biased towards texture; increasing shape bias improves accuracy and robustness[J]. arXiv preprint arXiv:1811.12231, 2018.

2. Section 3.2.3 does not show the relation between VQ and textural features.
3. The clinical motivation is not clear. For X-ray and Ultrasound images, both adopting transformer methods and using textural features are not explained from the clinical views.

**Questions:**

See the above weaknesses.

---

### Official Review · Reviewer_yqHn · 2023-11-09

**Soundness:** 2 fair
**Presentation:** 2 fair
**Contribution:** 1 poor
**Rating:** 1
**Confidence:** 4

**Summary:**

This paper proposes a texture transformer network (TxTN) by integrating three texture layers to enhance the discriminative capability of Vision Transformer (ViT) for medical image analysis. Three texture patterns (GLCM, VQ and Laplacian) are embedded into the design of ViT architecture to address its major shortcomings, including topological destruction, the loss of geometrical information and the lack of global characteristics. Then, the comparison experiments verify its effectiveness evaluated on two public medical datasets.

**Strengths:**

1. This approach tries to improve the performance of the vision transformer, which is essential for further medical image analysis.
2. Experimental results show the value of the proposed method on two public medical datasets compared to some baselines.

**Weaknesses:**

1. The first weakness is the lack of sufficient comparison experiments. This paper's chosen baselines (MobileNet, SWIN and VIT) are designed for general image classification in CV. The authors should select other specific baseline algorithms on breast and pneumonia datasets.
2. This paper could add more comparison visualization figures among different approaches.
3. The proposed texture transformer network lacks the novelty. The proposed pattern& texture module could be seen as one of the pre-processing operations.

**Questions:**

1. Please give the evidence to support the claim: "This inspirational idea stems from one important insight into the architecture of ViT and its major shortcomings, including topological destruction, the loss of geometrical information and the lack of global characteristics."
2. Why are all images resized to 72*72 in the experiments? It is too small.

---

### Meta-Review · Area_Chair_JtPr · 2023-11-30

**Metareview:**

This paper aims to explore texture patterns and features (such as gray level co-occurrence matrix and histogram) to improve transformer neural network for medical image analysis. Overall, the paper is weak in novelty and experimental validation. There is a consensus among the reviewers on the weakness of the paper and the authors did not provide any response the reviewers' comments. The authors need to improve the paper in both novelty and experimental validation.

**Justification For Why Not Higher Score:**

NA

**Justification For Why Not Lower Score:**

NA

---

### Decision · Program_Chairs · 2024-01-16

Reject